# NestedFP: High-Performance, Memory-Efficient Dual-Precision Floating Point Support for LLMs

**Haeun Lee, Omin Kwon, Yeonhong Park**[*]**, Jae W. Lee**[*]
Department of Computer Science and Engineering
Seoul National University
{haeunlee, om0127, ilil96 jaewlee}@snu.ac.kr

## Abstract

Meeting service-level objectives (SLOs) in Large Language Models (LLMs) serving is critical, but managing the high variability in load presents a significant challenge. Recent advancements in FP8 inference, backed by native hardware support, offer a potential solution: executing FP16 models by default, while switching to FP8 models during sudden load surges to achieve higher throughput at the cost of a slight quality degradation. Although this approach facilitates effective SLO management, it introduces additional memory overhead due to storing two versions of the same model. In response, this paper proposes NestedFP, an LLM serving technique that supports both FP16 and FP8 models in a memory-efficient manner by overlaying FP8 parameters onto FP16 parameters, allowing both models to share the same FP16 memory footprint. By leveraging a compact data format for the overlay and a specialized GEMM kernel optimized for this format, NestedFP ensures minimal degradation in both model quality and inference throughput across both FP8 and FP16 modes. NestedFP provides a flexible platform for dynamic, SLO-aware precision selection. The code is available at https://github.com/SNU-ARC/NestedFP.

## 1 Introduction

Large Language Models (LLMs) have emerged as foundational components in modern AI systems, enabling a wide range of applications such as virtual assistants, text and code generation, and multi-modal tasks that integrate textual and visual information [1, 7, 21]. Their broad applicability and strong performance have driven widespread adoption, leading to substantial user demand. For example, OpenAI's production-scale models process roughly 100 billion words per day [12] and production query rates frequently surpass 200 requests per second [17, 37].

To accommodate such large-scale applications, research efforts have focused not only on improving serving throughput [16, 38] but also on meeting service-level objectives (SLOs) [2, 13, 29]. One of the key challenges in achieving SLOs for LLM serving lies in handling dynamic load. Request rates, as well as input and output sequence lengths, can fluctuate significantly over short time intervals. Simply overprovisioning compute resources may mitigate this issue but results in substantial system cost inefficiency.

As a promising alternative, we explore an approach we call dual-precision LLM serving, which utilizes FP8 together with FP16, the de facto standard number format for weight parameters. While FP8 may incur a slight degradation in model quality, it offers up to 2× higher peak throughput compared to FP16 [22, 27, 28]. By operating in FP16 mode under normal conditions and switching to FP8 mode only during sudden load surges, we can effectively manage dynamic workloads without resource overprovisioning. This approach achieves a better balance between SLO attainment and service quality than relying solely on either FP16 or FP8.

---

[*]Corresponding authors

39th Conference on Neural Information Processing Systems (NeurIPS 2025).

Co-deploying models in both formats, however, is non-trivial. Simply keeping both in memory is infeasible due to excessive memory capacity overhead. Storing only FP16 weights and quantizing them to FP8 on the fly when needed results in highly suboptimal FP8 throughput. Therefore, an effective solution is needed to support dual-precision LLM serving without incurring memory overhead or throughput degradation.

In this paper, we present NestedFP, a technique that enables efficient dynamic selection between FP8 and FP16 precision from a single unified model representation. NestedFP decomposes each 16-bit weight tensor into two 8-bit components, allowing FP8 inference without additional memory overhead or throughput degradation. Since FP16 mode requires reconstructing 16-bit values by merging the two 8-bit components in this approach, we develop a custom General Matrix Multiplication (GEMM) kernel, built on the CUTLASS library [24], that performs accurate on-the-fly FP16 reconstruction during kernel execution. According to our evaluation, NestedFP enables high-quality, dual-precision LLM serving efficiently: FP8 models generated from decomposed FP16 weights match the quality of existing FP8 quantization methods, and our FP16 GEMM kernel incurs only a 6.38% average performance gap relative to the CUTLASS baseline, which translates to 4.98% end-to-end overhead across models.

Our contributions are summarized as follows:

- We propose dual-precision LLM serving as an effective strategy to handle dynamic load fluctuations.
- We introduce a novel data format that supports both high-quality FP16 and FP8 inference in a memory-efficient manner.
- We develop a custom FP16 GEMM kernel for this data format, which performs on-the-fly FP16 value reconstruction during execution.
- We demonstrate that NestedFP enables high-quality dual-precision LLM serving with no memory overhead and high throughput, significantly improving SLO attainment compared to FP16-only baselines.

## 2   Background

### 2.1   Floating Point Representation

A floating point format with $x$ exponent bits and $y$ mantissa bits is denoted as E$x$M$y$. The exponent width $x$ determines the dynamic range of representable values, while the mantissa width $y$ governs numerical precision. A floating point value $X_{FP}$ in this format can be expressed as shown in Equation 1, where $S$ is the sign bit, $M_j$ are the mantissa bits, $E$ is the unsigned integer formed by the exponent bits, and $b$ is the exponent bias.

$$X_{FP} = (-1)^S \cdot \left(1 + \sum_{j=1}^{y} M_j \cdot 2^{-j}\right) \cdot 2^{E-b}, \quad E \in \{0, 1, \ldots, 2^x - 1\} \tag{1}$$

Commonly used 16-bit formats include FP16 (E5M10) and BF16 (E8M7), while prevalent 8-bit formats include E4M3 and E5M2. These formats are natively supported by many modern hardware platforms and have become standard choices in low-precision machine learning computations [14].

### 2.2   FP8 Quantization for LLMs

**Rise of FP8 Quantization for LLMs.** While FP16 has become the de facto standard for LLM serving, 8-bit floating-point formats (FP8) are gaining traction for scenarios demanding even greater efficiency. This trend is driven by two main factors: (1) the representational advantages of floating-point formats, and (2) the increasing hardware support for FP8 arithmetic.

First, floating-point quantization is generally better suited for LLMs than integer quantization due to its wider dynamic range. Unlike in computer vision tasks—where integer quantization is prevalent—LLMs often exhibit activation outliers amplified by operations such as LayerNorm [28]. These outliers can significantly degrade model performance if not properly managed [14]. Although recent work has attempted to mitigate this issue in integer quantization through static outlier-aware techniques [28, 36], these methods still underperform compared to floating-point quantization, which inherently better accommodates long-tailed distributions and high dynamic ranges [20, 22, 40].

Second, FP8 support is rapidly being adopted by modern hardware accelerators, including NVIDIA Hopper GPUs and Intel Gaudi HPUs [14]. Leveraging this support, FP8 GEMM operations can achieve up to $2\times$ speedup compared to their FP16 counterparts. This growing availability is accelerating the adoption of floating-point quantization—particularly FP8—further highlighting its relevance for efficient LLM deployment.

**E4M3 Format.** Among the two widely adopted FP8 formats—E4M3 and E5M2—existing studies have consistently shown that E4M3 yields higher inference accuracy for Transformer-based language models [22, 28, 31]. However, due to its reduced dynamic range compared to FP16, the effectiveness of E4M3 critically depends on the application of appropriate scaling strategies [15, 22, 31]. Scaling is applied to align the dynamic range of tensor values with the numerical bounds of the E4M3 format. Scaling factors may be determined statically—based on heuristics such as the absolute maximum, percentile clipping, or mean squared error (MSE) minimization [22]—or computed dynamically during runtime. Weight tensors are typically scaled statically on a per-channel basis [20, 28, 40], most commonly using the absolute maximum value. Activation tensors, by contrast, may be scaled offline on a per-tensor basis [20, 28, 40] or dynamically on a per-token basis during inference. While the latter approach can improve accuracy, it incurs additional runtime cost due to the overhead of computing scaling factors on-the-fly [14, 27].

**FP8 Results.** While FP8 inference offers significant throughput gains, it comes with the potential caveat of quality degradation due to the inevitable loss of information associated with reduced precision [39]. Table 1 presents a comparison between FP8 and FP16 across various models and benchmarks. Specifically, for FP8 quantization, we use the E4M3 format with per-token activation and per-channel weight quantization for all linear layers, using the vLLM framework [33]. A consistent accuracy degradation is observed in FP8 models, which clearly shows the trade-off between throughput and model quality in FP8 quantization.

| | Llama 3.1 8B | | | Mistral Nemo | | | Phi-4 | | | Mistral Small | | |
|---|---|---|---|---|---|---|---|---|---|---|---|---|
| | FP16 | FP8 | $\Delta$ | FP16 | FP8 | $\Delta$ | FP16 | FP8 | $\Delta$ | FP16 | FP8 | $\Delta$ |
| Minerva Math | 18.2 | 17.6 | -0.60 | 17.2 | 16.5 | -0.72 | 42.7 | 42.9 | +0.14 | 35.4 | 34.2 | -1.14 |
| MMLU Pro | 33.3 | 32.8 | -0.49 | 35.4 | 35.3 | -0.06 | 53.1 | 52.7 | -0.36 | 54.3 | 53.6 | -0.77 |
| BBH | 39.5 | 38.6 | -0.86 | 41.6 | 40.6 | -0.97 | 27.5 | 27.1 | -0.40 | 52.4 | 50.8 | -1.62 |

Table 1: Downstream task accuracy for FP8 quantized models relative to FP16 models.

## 3 Motivation

### 3.1 Challenge: Dynamic Load in LLM Serving

LLM serving is characterized by substantial variability in request rates, as well as input prompt and output sequence lengths across requests [19, 30]. These variability is particularly problematic because the batch size scheduled at each iteration can fluctuate significantly, directly affecting the large deviations in Time To First Token (TTFT) and Time Per Output Token (TPOT). Such fluctuations occur in a highly dynamic and often unpredictable manner [29], making it difficult for service providers to consistently meet SLOs.

To quantify the extent of this variability, we analyze production traces from Microsoft Azure's LLM inference service [5]. Specifically, we measure how the request rate fluctuates over a one-hour interval (00:00–01:00 UTC, May 10, 2024, in the trace). Figure 1a presents the results. The request rate exhibits nearly a fivefold variation between the lowest and highest load periods. Moreover, as shown in the one-minute zoom-in view, these fluctuations occur at high temporal frequency, indicating second-level variability.

A classic approach to handling such load variations is auto-scaling, which adaptively adjusts computational resources (e.g., increasing or decreasing the number of GPUs in a cluster) [8]. However, auto-scaling is primarily effective in cloud environments with abundant resources and typically operates on minute-to-hour timescales [10]. Consequently, it is inadequate for LLM serving systems, which must respond to sharp, second-level load spikes and scale throughput within millisecond-level intervals to meet SLO requirements.

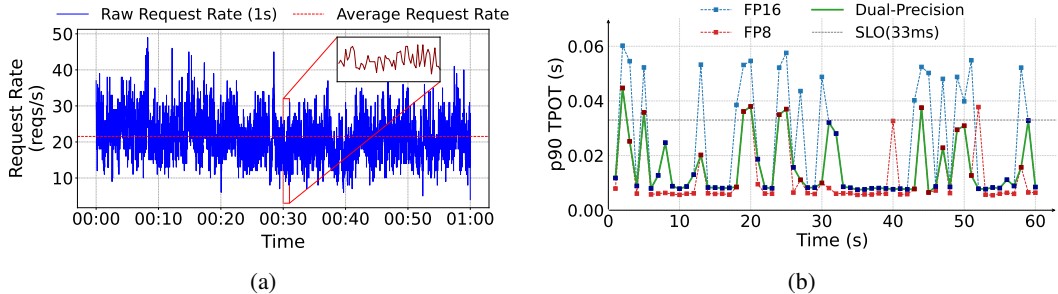

Figure 1: (a) Request rate fluctuations during the first hour of May 10, 2024, from the Azure LLM inference trace. The inset plot highlights 1-minute variability from the 30-minute mark. (b) Comparison of p90 TPOT across different model precisions, using the Azure LLM inference trace recorded on May 10, 2024.

## 3.2 Our Proposal: Dual-Precision LLM Serving

To address short-term fluctuations in resource demand during LLM serving, we propose a simple yet effective dual-precision strategy that dynamically switches between FP16 and FP8 modes. When the system load is low, FP16 is used to preserve maximum model quality. Under high load conditions, the system falls back to FP8 mode to increase throughput, thereby maintaining responsiveness. Although FP8 mode may incur slight accuracy degradation, it is often preferable to violating SLOs during periods of request surges.

Figure 1b presents the per-second p90 TPOT under three precision schemes: FP16, FP8, and the proposed dual-precision format, using the Llama 3.1 8B model on an H100 GPU with vLLM. The request patterns are extracted from a 60-second segment of Microsoft Azure's LLM inference trace [5], scaled down to 20% of the original load. The TPOT SLO threshold is set to 33 ms [6]. As shown, the FP16 model exhibits more frequent TPOT spikes that exceed the TPOT SLO threshold compared to the FP8 model. To further quantify the difference, we analyze the rate of SLO violations across the same trace: 35.8% of requests under FP16 exceeded the threshold, whereas FP8 reduced this to 17.3%. In short, the proposed dual-precision inference can achieve FP8-level SLO compliance while minimizing quality degradation.

## 3.3 How to Implement Dual-Precision LLM Serving?

To enable dynamic switching between FP16 and FP8 modes in LLM serving, two straightforward implementation strategies can be considered: separate storage and on-the-fly dequantization.

**Separate Storage.** One simple approach is to maintain both FP16 and FP8 models in memory and switch between them according to system load fluctuations. While this enables fast switching, it introduces a 50% memory overhead—a significant burden for LLM serving systems already constrained by model size. This memory capacity overhead not only limits the maximum deployable model size but also reduces the available memory for key-value (KV) caching, thereby restricting the number of concurrent requests and ultimately degrading overall serving throughput.

**On-the-fly Quantization.** An alternative approach is to store only the FP16 model and perform on-the-fly quantization to perform computation in FP8. Unlike the separate storage strategy, this approach avoids additional memory capacity overhead. However, its FP8 performance is suboptimal; although computation is carried out by FP8 units with high throughput, the full 16-bit weights must still be loaded from memory, effectively doubling memory traffic compared to native FP8 execution. As a result, memory bandwidth becomes a bottleneck, limiting the achievable throughput and diminishing much of FP8's advantage over the FP16 baseline.

These limitations highlight the need for a more sophisticated dual-precision serving mechanism—one that eliminates additional memory overhead while enabling efficient GEMM execution in both FP8 and FP16 modes.

## 4 NestedFP

**Overview.** Figure 2 illustrates the core concept of NestedFP, which stores only 16-bit weights in memory and extracts 8-bit weights from them when operating in FP8 mode. Specifically, NestedFP decomposes each FP16 parameter into two parts: the upper 8 bits and the lower 8 bits. With these two parts of the weight parameters stored as separate tensors, both parts are loaded in FP16 mode, whereas only the upper part is loaded in FP8 mode. This approach not only eliminates additional memory usage but also prevents memory traffic amplification—for example, fetching 16-bit weights when executing in 8-bit mode.

**Challenge 1: 8-bit Mode Accuracy.** One key challenge of this approach is ensuring that the 8-bit representation extracted from 16-bit parameters achieves sufficient accuracy. In other words, it must maintain performance comparable to existing FP8 quantization methods. Naive truncation (simply using the upper 8 bits of FP16 values) yields a representation similar to E5M2, which offers limited precision compared to the commonly preferred E4M3 format for LLM inference. Moreover, such truncation often performs worse due to indiscriminate rounding, further degrading accuracy. This motivates the need for a quantization-aware data format that enables the extraction of high-quality FP8 representations. Section 4.1 elaborates on this data format design.

**Challenge 2: 16-bit Mode Throughput.** Another key challenge is maintaining throughput in the 16-bit mode. The 8-bit mode in NestedFP trivially achieves high performance, as native FP8 GEMM kernels can be used directly after loading the upper-part tensor. However, the 16-bit mode is less straightforward: both the upper and lower-part tensors must be loaded and combined to reconstruct the original 16-bit weights. This requires a custom GEMM kernel capable of performing the reconstruction dynamically. Section 4.2 details this kernel design.

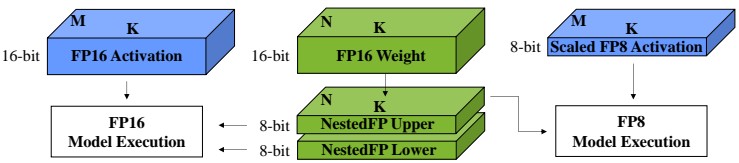

Figure 2: FP16 weight is decomposed into 8-bit upper and lower parts. Both parts are used for FP16 mode while only upper part is used for FP8 mode.

### 4.1 Compact yet Effective Data Format for Dual-Precision LLM

**Opportunities: Low-Entropy Exponent Bit in FP16 Models.** Figure 3a shows the the weight distributions of all FP16 linear layers across four different LLMs. The vast majority of weight values have absolute magnitudes less than or equal to 1.75, which means that their most significant exponent bits in FP16 are zero. Specifically, as shown in Figure 3b, three of the four evaluated models exhibit this trait across all layers. The sole exception, Phi-4, contains some layers that deviate from this pattern, but these account for only 8.75% of its layers. This strong tendency suggests an opportunity for lossless exponent mapping from FP16 to the FP8 E4M3 format, which also uses 4 exponent bits.

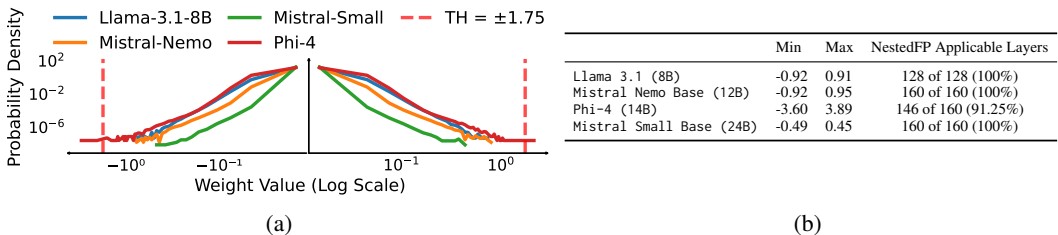

| | Min | Max | NestedFP Applicable Layers |
|---|---|---|---|
| Llama 3.1 (8B) | -0.92 | 0.91 | 128 of 128 (100%) |
| Mistral Nemo Base (12B) | -0.92 | 0.95 | 160 of 160 (100%) |
| Phi-4 (14B) | -3.60 | 3.89 | 146 of 160 (91.25%) |
| Mistral Small Base (24B) | -0.49 | 0.45 | 160 of 160 (100%) |

| (a) | (b) |
|---|---|

Figure 3: (a) Weight distributions of linear layers across different models. (b) Per-model weight range (min/max) and the proportion of linear layers where NestedFP can be applied.

**Our Proposal.** Figure 4a illustrates how NestedFP decomposes an FP16 weight such that the upper 8 bits can serve as a high-quality E4M3 representation by leveraging an unused exponent bit in FP16. First, the lower 8 bits directly inherit the lower 8 bits of the FP16 mantissa. The upper 8 bits, which

correspond to the E4M3 representation, are constructed as follows. The sign bit is directly inherited from FP16. The next four exponent bits are also copied from FP16, excluding the most significant exponent bit, which is assumed to be unused (or zero). Finally, 3 most significant mantissa (i.e., bits M[1:3]) bits are obtained and constructed value is applied rounding using a round-to-nearest-even policy. Specifically, the 7 least significant bits of the FP16 mantissa (i.e., bits M[4:10]) are examined to determine whether the value lies above or below the midpoint (64). If the value is greater than 64, it is rounded up; if it equals 64, it is rounded up only when the least significant bit of the resulting 3-bit mantissa (M3) is 1; otherwise, it is rounded down. This decomposition process is performed offline prior to model execution.

In this scheme, removing most significant exponent bit effectively acts as multiplying $2^8$ between E4M3 and FP16 value. In other words, NestedFP effectively applies E4M3 quantization with a fixed scaling factor of $2^8$.

**FP16 Reconstruction.** Figure 4b illustrates how NestedFP reconstructs FP16 weights at runtime. Essentially, this process merges the upper and lower 8-bit parts, but not through a simple concatenation. The rounding decision made during decomposition must be reversed by checking an implicit checksum; if the LSB of the upper 8 bits differs from the MSB of the lower 8 bits, a rounding-down correction is applied before recombining the exponent and mantissa bits into a valid FP16 value. This process ensures precise, lossless recovery of the original FP16 weights.

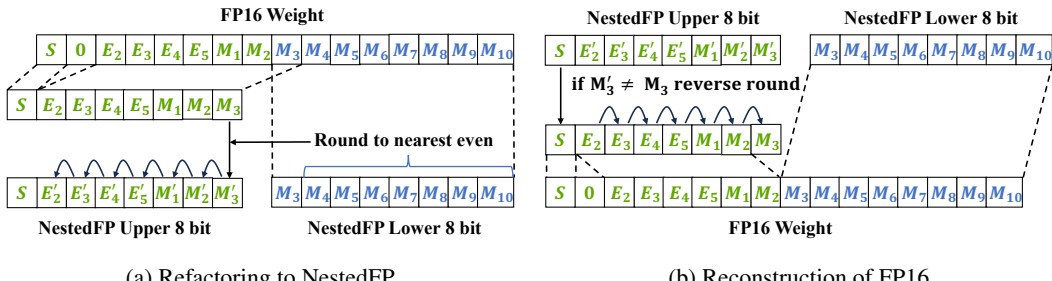

(a) Refactoring to NestedFP  (b) Reconstruction of FP16

Figure 4: (a) Decomposition of FP16 weight values into two 8-bit parts (offline). (b) Reconstruction of FP16 weight values (online).

**Handling Exception Layers.** For layers containing weight values with magnitudes exceeding 1.75 (i.e., where the MSB of the exponent is possibly non-zero), we retain them in FP16 and always perform FP16 computation. This may reduce the benefits of dual-precision LLM serving, as the throughput advantage of FP8 mode becomes less pronounced. However, it is important to recall that, as shown in our analysis of popular models (Figure 3), such layers are rare and therefore have minimal impact on the overall effectiveness of the dual-precision scheme.

## 4.2 FP16 GEMM Kernel with On-the-fly Reconstruction

Figure 5 illustrates the high-level execution flow of the NestedFP FP16 GEMM kernel compared to a standard FP16 GEMM kernel. We assume an NVIDIA GPU environment, where the CUTLASS implementation serves as the standard FP16 GEMM kernel, upon which our kernel is also built. In the standard FP16 GEMM kernel, weights are loaded into shared memory, transferred to registers, and directly consumed by compute units (e.g., tensor cores in NVIDIA GPUs) without any transformation. In contrast, the FP16 GEMM in NestedFP requires a reconstruction step before tensor core execution; weights are stored as two 8-bit tensors, which must be loaded into shared memory, copied to registers, and then reconstructed into full FP16 values. Only after this reconstruction can the data be used by the tensor cores.

Not to make this additional step of reconstruction introduce a bottleneck, we present two key optimizations: (1) merged bitwise operations and (2) a three-stage pipeline.

**Efficient FP16 Reconstruction via Merged Bitwise Operations.** FP16 reconstruction involves a large number of 8-bit bitwise operations. To reduce the associated overhead, we fuse four 8-bit operations into a single 32-bit instruction, following the approach in [26, 35]. Algorithm 1 illustrates how FP16 reconstruction is performed using merged bitwise operations. Conceptually, the process iterates over the weight tensor, where each iteration processes four elements. In each iteration, four

8-bit values are loaded from the upper and lower tensors ($W_{\text{upper}}$ and $W_{\text{lower}}$) into 32-bit registers u and l, respectively (Line 5–6). Then, the sign bits are extracted from u and stored in another register s (Line 7). Next, the algorithm checks whether rounding has occurred by examining l (Line 8); if rounding is detected, the effect is reversed, thereby recovering the exponent and mantissa parts of the upper 8 bits (Line 9). Then, the original upper 8 bits, u_orig, are fully reconstructed by concatenating them with s (Line 10). Finally, u_orig and l are fused using NVIDIA's __byte_perm instruction (lines 11–12), completing the reconstruction. Note that all the above operations are simultaneously applied to four 8-bit values through a 32-bit instruction, effectively reducing the total number of bitwise operations required.

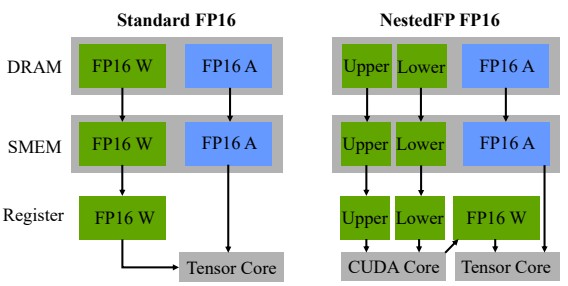

Figure 5: FP16 execution flow comparison.

**Algorithm 1** FP16 Weight Reconstruction

```
1  W_upper, W_lower: upper and lower 8-bit tensor
2  W: reconstructed FP16 weight tensor
3  N: total number of elements in the weight tensor
4  for i = 0 to N step 4 do
5      uint32_t u = W_upper[i : i + 4]
6      uint32_t l = W_lower[i : i + 4]
7      uint32_t s = u & 0x80808080
8      uint32_t sub = (l & 0x80808080) ≫ 7
9      uint32_t u_orig = (((u − sub) ≫ 1) & 0x3f3f3f3f)
10     u_orig = u_orig | s
11     W[i : i + 2] = __byte_perm(u_orig, l, 0x1504)
12     W[i + 2 : i + 4] = __byte_perm(u_orig, l, 0x3726)
13 end for
```

**Three-Stage Pipeline.** The CUTLASS FP16 GEMM kernel, upon which the NestedFP FP16 kernel is built, employs a two-stage pipeline to overlap data transfer and computation. Specifically, the two stages correspond to (1) data movement between shared memory and registers and (2) Tensor Core execution, which are overlapped for high throughput. NestedFP extends this design into a three-stage pipeline to further hide reconstruction overheads. In particular, the three stages in NestedFP are: (1) shared-memory-to-register data transfer, (2) FP16 reconstruction, and (3) Tensor Core execution.

## 5   Evaluation

Through extensive experiments, we demonstrate that NestedFP is an effective solution for handling fluctuating loads in LLM serving by dynamically switching between FP16 and FP8 modes, as evidenced by the following three key results:

- The quality of FP8 model extracted from FP16 parameters in the NestedFP data format matches state-of-the-art FP8 quantization results. (Section 5.1)
- The FP16 mode of NestedFP achieves throughput comparable to baseline FP16 inference using the vanilla CUTLASS kernel, despite performing on-the-fly FP16 reconstruction. (Section 5.2 and Section 5.3)
- NestedFP significantly improves SLO attainment over FP16-only deployment, without introducing additional memory overhead. (Section 5.4)

### 5.1   FP8 Model Quality

**Methodology.** We evaluate the accuracy of the FP8 model in NestedFP against FP8 models quantized using commonly adopted configurations, specifically per-channel weight quantization. For activation quantization, we apply per-token quantization for both NestedFP and the baseline. The evaluation covers six models: Llama 3.1 8B, Mistral Nemo (12B), Phi-4 (14B), Mistral Small (24B), Llama 3.1 70B, and DeepSeek-R1-Distill-Llama-70B [3, 4, 9, 21, 23]. For benchmarks, we use three downstream tasks: Minerva Math, MMLU Pro, and BBH [18, 32, 34]. All models are integrated into the vLLM framework, and accuracy is measured using the LM Evaluation Harness [11]. As noted in Section 4.1, 8.75% of the linear layers in Phi-4 are not applicable to NestedFP and are therefore executed in FP16. More details on the experimental setup are provided in Appendix A.

**Results.** Table 2 presents the results. Across all models and benchmarks, the FP8 model in NestedFP achieves accuracy comparable to the baseline FP8 model, demonstrating the effectiveness of the new data format of NestedFP.

| Model | Size | Minerva Math | | MMLU Pro | | BBH | |
|---|---|---|---|---|---|---|---|
| | | FP8(B) | FP8(N) | FP8(B) | FP8(N) | FP8(B) | FP8(N) |
| Llama 3.1 | 8B | 17.6 | 17.2 | 32.8 | 33.2 | 38.6 | 39.0 |
| Mistral Nemo | 12B | 16.5 | 16.4 | 35.3 | 34.7 | 40.6 | 40.0 |
| Phi-4 | 14B | 42.9 | 42.9 | 52.7 | 53.3 | 27.1 | 28.7 |
| Mistral Small | 24B | 34.2 | 35.2 | 53.6 | 53.7 | 50.8 | 51.1 |
| Llama 3.1 | 70B | 35.7 | 34.6 | 48.1 | 47.4 | 49.1 | 49.7 |
| DeepSeek-R1-Distill-Llama | 70B | 46.0 | 45.5 | 49.2 | 49.4 | 53.2 | 53.4 |

Table 2: Accuracy of FP8 models in NestedFP across various downstream tasks. FP8(B) denotes the baseline model, and FP8(N) denotes NestedFP.

## 5.2 FP16 GEMM Kernel Performance

**Methodology.** We evaluate the FP16 GEMM kernel performance of NestedFP against the CUTLASS FP16 GEMM kernel, upon which ours is built, using an NVIDIA H100 GPU. Note that NestedFP does not require any custom kernel for FP8 GEMM; thus, we omit FP8 kernel evaluation. We consider four different weight matrix shapes: (N, K) = (28672, 4096), (28672, 5120), (35840, 5120), (65536, 5120). The $(M)$ dimension, which corresponds to the number of tokens processed in parallel, is varied from 32 to 2048 in increments of 32. For both kernels, we select the best configuration (e.g., tile size) for each GEMM shape through an exhaustive search over the design space. Details of the kernel search space are provided in Appendix D.2.

**Results.** Figure 6 shows the results. The NestedFP kernel consistently demonstrates performance comparable to the CUTLASS baseline across all GEMM shapes. Specifically, the average relative overheads for each weight matrix dimension are 5.89%, 6.33%, 4.59%, and 6.33%, respectively. These results indicate that the NestedFP kernel effectively minimizes the overhead introduced by on-the-fly FP16 reconstruction. Additional results for other GEMM shapes, which exhibit similar trends, as well as comparisons with cuBLAS FP16 kernels, are provided in Appendix B and Appendix D.1.

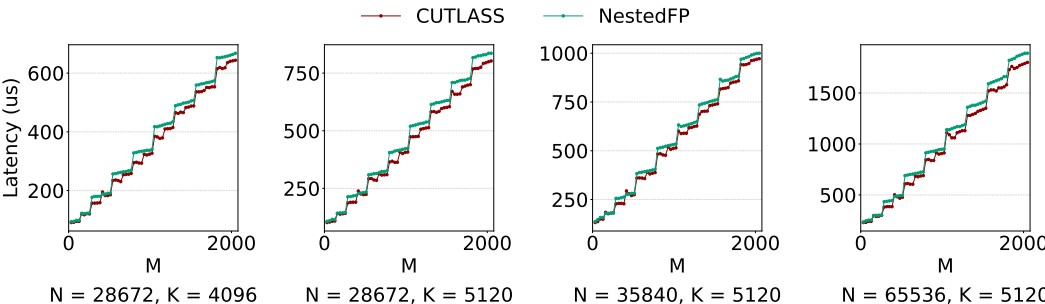

Figure 6: CUTLASS baseline and NestedFP kernel performance comparison.

## 5.3 FP16 Mode End-to-End Inference Performance

**Methodology.** We evaluate the end-to-end inference throughput of the FP16 mode of NestedFP by integrating it into the vLLM framework on a single NVIDIA H100 GPU. Specifically, we set the input and output sequence lengths to 1024 and 512, respectively, while varying the batch size from 32 to 512. More details on the experimental setup are provided in Appendix C. The results are compared against vanilla vLLM FP16 execution, which relies on PyTorch for invoking GEMM kernels. The FP8-mode throughput of NestedFP is excluded from the evaluation, as it does not require a separate kernel.

**Results.** Figure 7 presents the results. NestedFP achieves FP16-mode throughput highly comparable to the baseline, with only minor overheads—5.04% (Llama 3.1 8B), 5.56% (Mistral Nemo), 4.76% (Phi-4), and 4.56% (Mistral Small) on average across batch sizes. The end-to-end throughput degradation is even lower than the kernel-level overheads reported in Section 5.2, since non-GEMM components also contribute to total execution time. Additional end-to-end throughput results are provided in Appendix C.

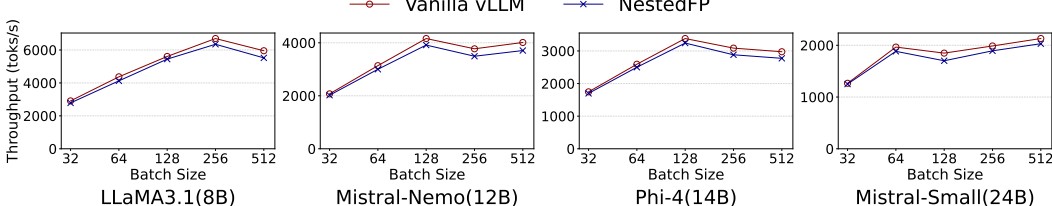

Figure 7: FP16 inference throughput comparison of four models between vanilla vLLM and NestedFP.

## 5.4 Impact on SLO Attainment

**Methodology.** In this section, we present a case study demonstrating how NestedFP can improve SLO attainment. We implement a simple load-aware mode switching policy for NestedFP that, in each iteration, selects FP8 mode when the number of tokens to process exceeds a predefined threshold (1024 in this experiment); otherwise, it selects FP16 mode. The evaluation is conducted using Llama 3.1–70B on a cluster equipped with four NVIDIA H100 GPUs. As input, we use 1,000 requests sampled from an Azure LLM inference trace while varying the request rate. We measure p90 TTFT, p90 TPOT, and SLO attainment under two criteria: loose and tight. The loose SLO is defined as $\leq 10\times$ the average single-request latency, and the tight SLO is defined as $\leq 5\times$ the average single-request latency, following the methodology of a previous study [13].

**Results.** Table 3 shows the results. NestedFP consistently outperforms the FP16-only baseline across all request rates in terms of TTFT, TPOT, and SLO attainment, with particularly pronounced improvements under high request rates. Note that this experiment employs a simple, straightforward switching policy, leaving substantial room for further optimization by integrating NestedFP with more sophisticated policies—such as those considering KV-cache utilization or incorporating load prediction mechanisms. Nevertheless, these results confirm that NestedFP provides an efficient dual-precision mechanism that can significantly enhance SLO attainment in LLM serving systems.

| Request Rate | Method | Tight SLO (%) | Loose SLO (%) | p90 TTFT (s) | p90 TPOT (ms) |
|---|---|---|---|---|---|
| 7.47 req/s | FP16 | 0.7 | 14.1 | 9.9893 | 162.2 |
| | NestedFP | 72.4 | 90.0 | 1.9518 | 118.9 |
| 5.03 req/s | FP16 | 71.6 | 93.6 | 1.6449 | 151.3 |
| | NestedFP | 94.6 | 99.9 | 0.9679 | 86.8 |
| 3.79 req/s | FP16 | 83.1 | 96.9 | 1.2404 | 109.2 |
| | NestedFP | 96.6 | 100 | 0.8168 | 60.1 |
| 1.99 req/s | FP16 | 90.4 | 99.4 | 0.8915 | 64.9 |
| | NestedFP | 98.4 | 100 | 0.6116 | 47.7 |

Table 3: SLO attainment and p90 TTFT/TPOT of Llama-3.1-70B on 4×H100 under varying loads.

## 6 Conclusion

We introduce NestedFP, a memory-efficient framework that supports both FP16 and FP8 precisions for LLM inference in a memory-efficient way. NestedFP introduces a new data format that enables direct extraction of FP8 weights from FP16 weights without additional memory overhead. NestedFP also incorporates a custom GEMM kernel optimized for this format to ensure efficient computation. As a result, NestedFP enables dynamic switching between FP16 and FP8 modes with the memory footprint of FP16, without noticeable degradation in either accuracy or inference throughput, providing an effective means to improve SLO attainment in LLM serving.

## 7 Acknowledgments

This work was supported by the National Research Foundation of Korea (NRF) grant funded by the Korea Government (MSIT) (RS-2024-00340008) and Institute of Information & Communications Technology Planning & Evaluation (IITP) under the artificial intelligence semiconductor support program (IITP-2023-RS-2023-00256081), funded by the Korea Government (MSIT).

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

## A   Evaluation Details

Experiments were conducted on NVIDIA H100 GPU with vLLM 0.8.5 (V1 engine) and PyTorch 2.7.0+cu12.6. For accuracy assessment, we use the Math Verify metric on Minerva Math, the Open LLM Leaderboard configuration for MMLU Pro, and a zero-shot setting for BBH. The $M$ dimension is padded to multiples of $T_m$ only for kernel microbenchmarks and is disabled in the vLLM integration.

## B   Extended Kernel Evaluation

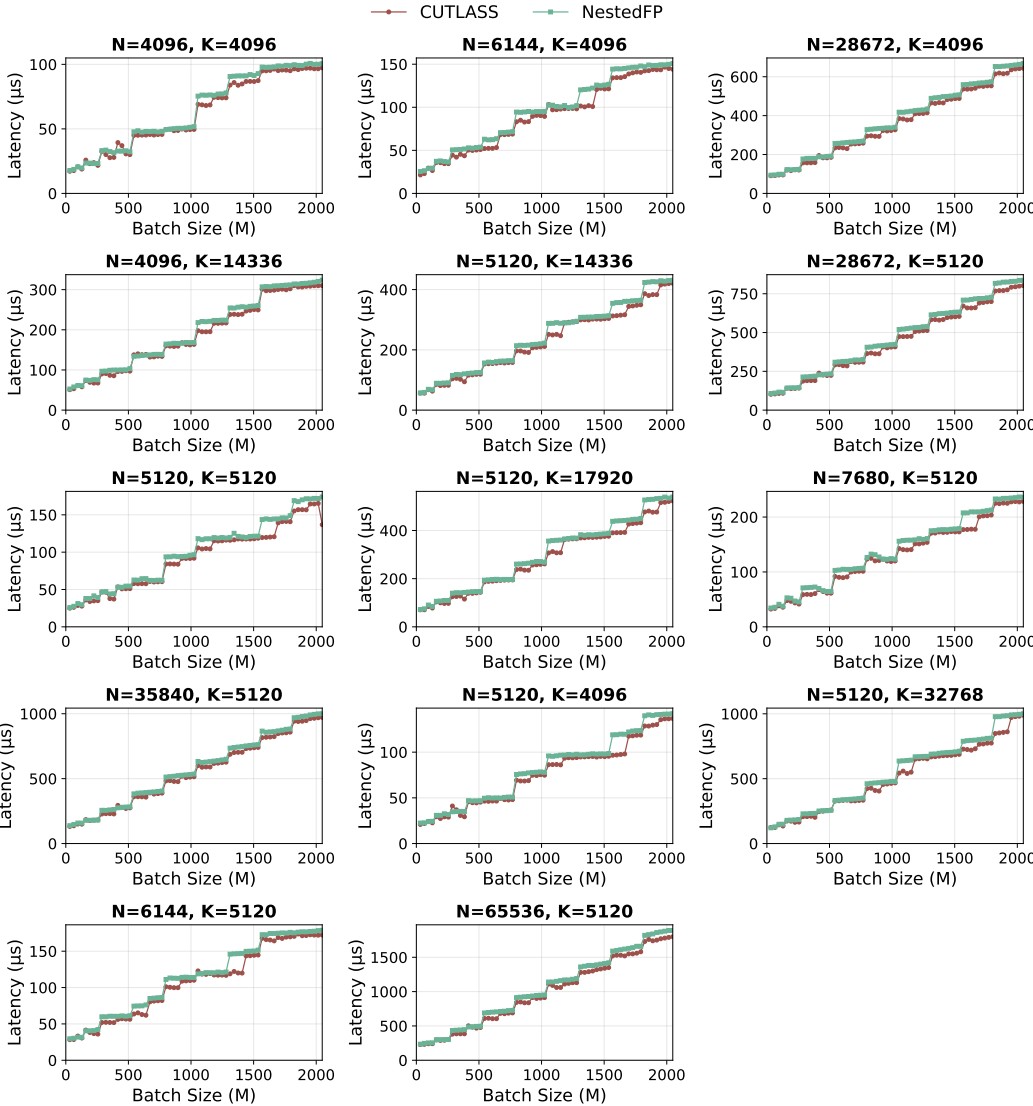

Figure 8: Comparison of kernel performance between the CUTLASS baseline and NestedFP.

Figure 8 compares NestedFP with the CUTLASS FP16 baseline across 14 $(N, K)$ GEMM shapes for four models: Llama 3.1 8B, Mistral Nemo, Phi-4, and Mistral Small. We vary $M$ from 32 to 2048 in increments of 32. Overall, NestedFP incurs a moderate overhead, with an average of 6.38% relative to the FP16 baseline.

# C    Extended End-to-End Throughput Evaluation

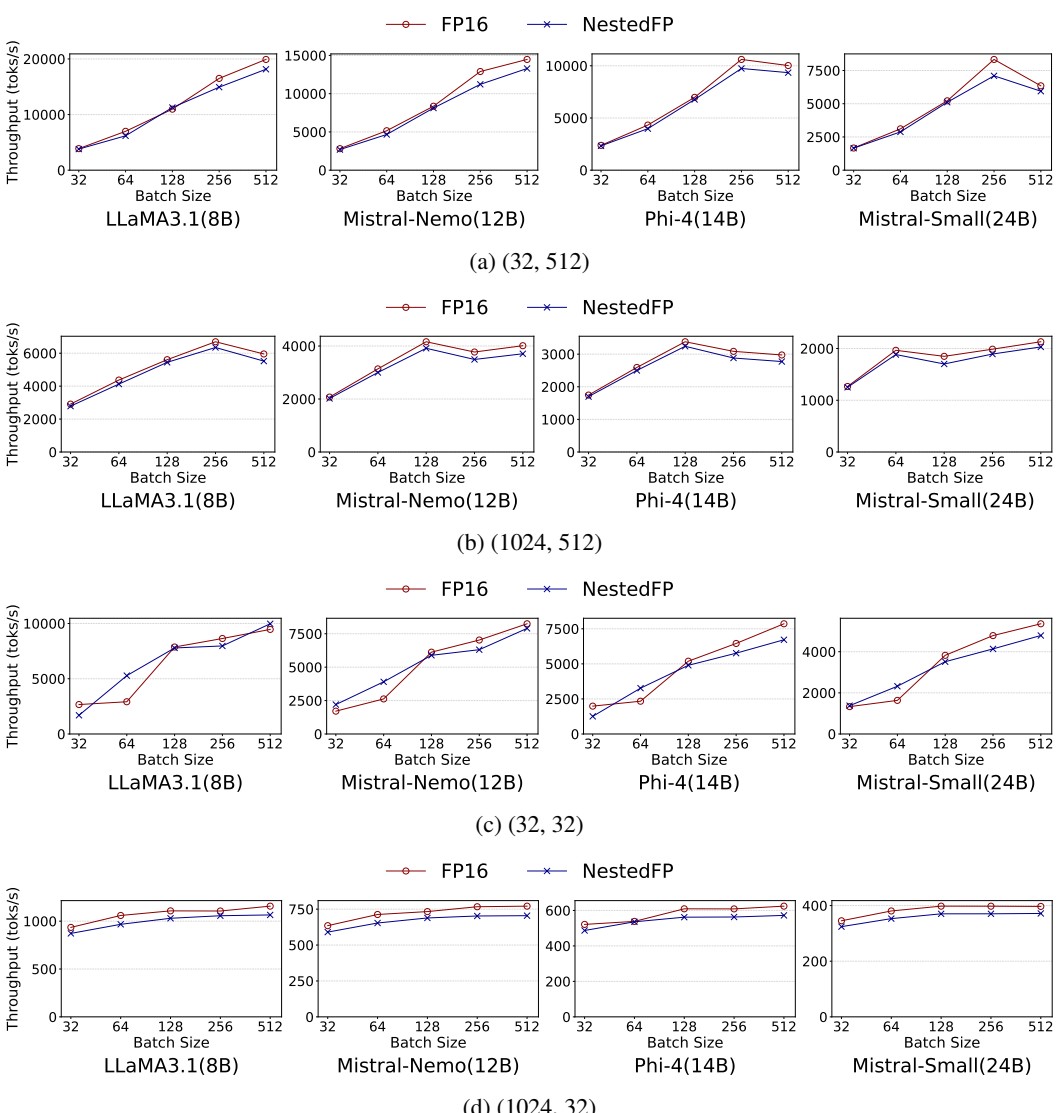

Figure 9: FP16 inference throughput comparison across four models under different input/output token configurations.

Figure 9 reports end-to-end throughput for (input, output) token pairs of (32, 512), (1024, 512), (32, 32), and (1024, 32). Across all settings, NestedFP achieves throughput comparable to the FP16 baseline. Chunked prefill was enabled, and the maximum number of batched tokens was set to 8192.

# D   CUTLASS FP16 GEMM Kernel

## D.1   CUTLASS Baseline and PyTorch Kernel Comparison

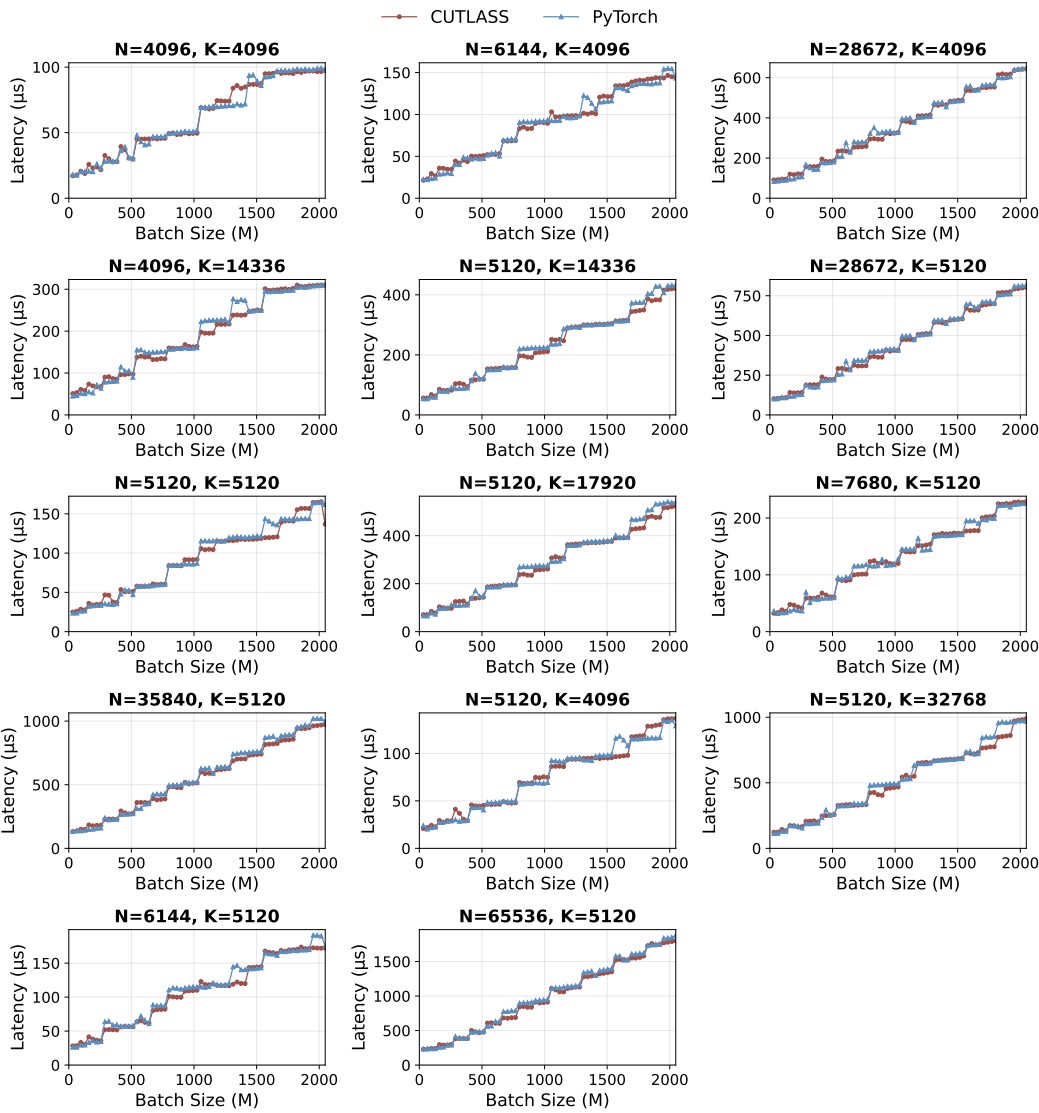

Figure 10: Comparison of kernel performance between CUTLASS baseline and PyTorch.

Figure 10 compares kernel performance between the CUTLASS FP16 baseline and PyTorch 2.6.0 (CUDA 12.4). We evaluate 14 $(N, K)$ GEMM shapes from four models: Llama 3.1 8B, Mistral Nemo, Phi-4, and Mistral Small, sweeping $M$ from 32 to 2048 in increments of 32. Overall, the CUTLASS baseline achieves performance comparable to PyTorch.

### D.2 Kernel Search Space

We build upon CUTLASS 3.6.0. In CUTLASS, four warps are grouped into a warp group. We evaluate two variants: a non-cooperative kernel (one warp group for TMA and one for MMA) and a cooperative kernel (one warp group for TMA and two for MMA). For non-cooperative kernels, we grid-search tile dimensions with $T_m \in \{16, 32, 64, 128, 256\}$, $T_n \in \{64, 128, 256\}$, and $T_k \in \{64, 128, 256\}$. For cooperative kernels, we vary only $T_n$, setting $T_n \in \{128, 256\}$. We exclude configurations that fail to compile. Non-cooperative kernels use a cluster shape of $(1, 1, 1)$, whereas cooperative kernels use $(2, 1, 1)$. We use the persistent scheduler for non-cooperative kernels, and consider both the persistent and Stream-K [25] schedulers for cooperative kernels.

## E  Extended Applicability Analysis

| Model | # Layers | # Applicable | Ratio (%) |
|---|---|---|---|
| CodeLlama 7B | 224 | 223 | 99.6 |
| CodeLlama 13B | 280 | 277 | 98.9 |
| Gemma 3 4B | 563 | 429 | 76.2 |
| Gemma 3 12B | 661 | 527 | 79.7 |
| Gemma 3 27B | 759 | 625 | 82.3 |
| Llama 3.1 8B | 224 | 224 | 100.0 |
| Llama 3.1 70B | 560 | 523 | 93.4 |
| Mistral Nemo 12B | 280 | 280 | 100.0 |
| Mistral Small 24B | 280 | 280 | 100.0 |
| Phi-3.5 Mini | 128 | 112 | 87.5 |
| Phi-4 14B | 160 | 146 | 91.2 |
| Qwen 3 8B | 252 | 249 | 98.8 |
| Qwen 3 14B | 280 | 278 | 99.3 |
| Qwen 3 32B | 448 | 438 | 97.8 |

Table 4: Layer-wise applicability of NestedFP across models.

Table 4 presents an applicability analysis across a broader range of models. We report the total number of layers, the number of applicable layers satisfying the weight-magnitude constraint ($|w| \le 1.75$), and the corresponding ratio. Overall, most models exhibit high applicability across layers.

