# OpenReview forum: "NestedFP: High-Performance, Memory-Efficient Dual-Precision Floating Point Support for LLMs"
_NeurIPS.cc/2025/Conference — NeurIPS 2025 poster_

### Official Review · Reviewer_B4rE · 2025-06-11

**Clarity:** 3
**Significance:** 2
**Originality:** 2
**Rating:** 4
**Confidence:** 4

**Summary:**

The paper presents NestedFP, a dual-precision weight representation scheme that enables both FP16 and FP8 execution from a single 16-bit model representation, avoiding additional memory overhead. It proposes a structured decomposition of FP16 weights into two 8-bit components, allowing FP8 inference with high throughput and FP16-level accuracy when needed. A custom CUTLASS-based FP16 GEMM kernel is developed to reconstruct full-precision weights on-the-fly. Integration with the vLLM serving framework demonstrates up to 1.55× throughput improvements in FP8 mode and minimal overhead (~3.9%) in FP16 mode. The paper provides a new perspective for LLM serving with dynamic precision adaptation based on system load and service-level objectives (SLOs).

**Questions:**

1. The proposed method sets a fixed scaling factor for FP8 quantization. Will this limit the flexibility of FP8 quantization? Table 2 partly answers this question. But do you have more ablations for this design – particularly the PPL (perplexity) of the LLM?

2. What are the FP16 accuracy / PPL numbers in table 2?

3. What is the baseline FP16 latency in Figure 7 (b)?

**Ethical Concerns:**

["NO or VERY MINOR ethics concerns only"]

**Final Justification:**

The authors' rebuttal has addressed most of my concerns. And I have raised my score accordingly.

**Limitations:**

yes

**Paper Formatting Concerns:**

N/A.

**Quality:**

3

**Strengths And Weaknesses:**

[Strengths]
1. The paper presents a new perspective for LLM serving with dynamic precision adaptation based on system load and service-level objectives (SLOs).
2. The paper provides evaluation across different LLMs such as LLaMA 3.1 8B, Mistral Nemo, Phi-4, and Mistral Small. It also introduces system-level optimizations such as CUTLASS-based GEMM kernel, SIMT pack instructions, and integration to serving framework vLLM.


[Weaknesses]
1. The motivation of the proposed method is not strong enough. The FP8 quantization already achieves very decent accuracy for LLMs, particularly with post-training quantization techniques such as SmoothQuant and even QAT (quantization-aware training) methods. The motivation to switch between FP8 and FP16 precision is not convincing enough.
2. Some of the claims/experiments in the paper need further clarification (see questions).

---

> ### Author Rebuttal · Authors · 2025-07-31
>
> Thank you for your thoughtful review and helpful questions.
>
> 1. **Doesn’t the SOTA FP8 model work as well as the FP16 model? Why keep both?**
> It is true that state-of-the-art methods such as SmoothQuant have advanced FP8 quantization to the point where it appears nearly lossless on some popular benchmarks. However, recent studies have shown that, when evaluated more thoroughly across a broader range of tasks, even the best-performing 8-bit models still significantly underperform compared to their FP16 counterparts [1]. As an example, human evaluation results from LLM Arena [2] highlight that the 8-bit version of Llama-3.1-405B-Instruct (FP8) exhibits noticeable performance degradation compared to its FP16 counterpart, particularly in tasks involving coding (score: 1277 vs. 1293) and long-context queries (1275 vs. 1282). Note, that Llama-3.1-405B-Instruct (FP8) used SmoothQuant for FP8 quantization [3]. This highlights the need to retain FP16 alongside FP8 to balance inference quality with SLO constraints, as FP8 may still introduce non-negligible errors in some cases.
> &nbsp;
> [1] Tianyi Zhang, Yang Sui, Shaochen Zhong, Vipin Chaudhary, Xia Hu, and Anshumali Shrivastava. 70% size, 100% accuracy: Lossless llm compression for efficient gpu inference via dynamic-length float, 2025.
> [2] https://x.com/lmarena_ai/status/1835760196758728898
> [3] https://arxiv.org/pdf/2407.21783
> &nbsp;
> 2. **PPL Evaluations**
> We report perplexity results on Wikitext2 (W2), C4, and PTB datasets across four models. The FP8 baseline—denoted as (B)—applies per-token activation and per-tensor weight quantization. Our method, NestedFP, is denoted as (N). Additionally, we report results for SmoothQuant, denoted as (S), which builds on (B) with additional activation smoothing.
> &nbsp;
> All quantization schemes were applied to the QKVO and MLP linear layers. For SmoothQuant, we followed the protocol from the original paper and computed activation statistics using the Pile training dataset. A grid search over α ∈ {0.5, 0.7, 0.9} was performed, and the best perplexity value was selected.
> &nbsp;
> Overall, FP8 quantization methods yield slightly higher perplexity than FP16, while SmoothQuant shows comparable performance to the FP8 baseline. The perplexity trends are similar with accuracy evaluations.
> &nbsp;
> | Model            | W2(FP16) | W2(B) | W2(N) | W2(S) | C4(FP16) | C4(B) | C4(N) | C4(S) | PTB(FP16) | PTB(B) | PTB(N) | PTB(S) |
> |------------------|-----------|--------|--------|--------|-----------|--------|--------|--------|-------------|---------|---------|---------|
> | Llama 3.1 8B     | 6.24      | 6.30   | 6.31   | 6.29   | 9.58      | 9.67   | 9.67   | 9.67   | 9.01        | 9.06    | 9.08    | 9.06    |
> | Mistral Nemo     | 5.74      | 5.80   | 5.81   | 5.78   | 9.57      | 9.65   | 9.67   | 9.63   | 8.66        | 8.73    | 8.75    | 8.70    |
> | Phi-4            | 6.46      | 6.47   | 6.49   | 6.48   | 12.07     | 12.09  | 12.11  | 12.10  | 9.32        | 9.34    | 9.35    | 9.34    |
> | Mistral Small    | 5.29      | 5.35   | 5.35   | 5.34   | 9.09      | 9.20   | 9.21   | 9.20   | 8.00        | 8.07    | 8.07    | 8.06    |
> &nbsp;
> 3. **FP16 results for models/tasks in Table 2**
> FP16 results are provided in Table 1. To improve clarity, we will include them in Table 2 in the future revision.
> &nbsp;
> 4. **Regarding Baseline FP16 Latency of Figure 7 (b)** Please refer to Appendix C for the FP16 baseline latency values. In Figure 7(b), “Lvl 3” corresponds to our optimized NestedFP16 kernel. Appendix C also includes a comparison graph for the GEMM shape M×5120×32768, illustrating the performance of both the FP16 baseline and NestedFP16.
> &nbsp;
> 5. **Regarding scaling factor** The reviewer is correct that NestedFP uses a fixed scaling factor when encoding the FP16 weight tensor into two FP8 components. This design choice enables exact reconstruction of the original FP16 weights, while maintaining compatibility with FP8 tensor operations.
> &nbsp;
> Importantly, this does not preclude additional scaling: weights may still be scaled as long as the values remain within the representable range of FP8 (E4M3), offering flexibility where needed.
> &nbsp;
> Crucially, our method places no restriction on how activations are quantized. While we adopt per-tensor activation quantization in our experiments, per-token quantization schemes—like those used in the FP8 baseline (B)—can be readily incorporated to further improve accuracy. In this sense, while weight quantization is fixed for exact reconstruction, activation quantization remains fully flexible.
> &nbsp;

---

> > ### Comment · Reviewer_B4rE · 2025-08-04
> >
> > Thank you for the response.
> >
> > Regarding the **PPL Evaluations**, why do you use per-tensor weight quantization instead of a per-channel weight quantization -- as the per-channel weight scaling is almost zero-overhead? Are there any results for the 'per-channel weight, per-token activation' FP8 quantization -- e.g., Wikitext2 perplexity with smoothquant?
> >
> > For the question regarding **motivation to switch between FP8 and FP16**, I have read the materials provided by the authors. The [twitter post](https://x.com/lmarena_ai/status/1835760196758728898) actually says that "In coding/longer queries, bf16 gets slightly higher score, but **remains within the confidence intervals**.... fp8 version could **closely match bf16 performance** while significantly reducing costs." Additionally, the [Llama-3 Tech Report](https://arxiv.org/pdf/2407.21783) cited in the authors' rebuttal also indicate that FP8 can well preserve the model's accuracy. For example, the caption of Figure 26 is "Our FP8 quantization approach has negligible impact on the model’s responses." Therefore, my concerns regarding the necessity to keep both FP8 and FP16 models still exists.

---

> ### Author Response · Authors · 2025-08-04
>
> We appreciate your thoughtful comments. Below we address your concerns in detail.
>
> 1. **Regarding PPL evaluations**
> We have additionally ran PPL experiments to show per-channel (per-row) wise weight quantization yields similar perplexity as per-tensor weight quantization. There is no significant difference. W2 denotes Wikitext2 dataset.
>
> | Model - Dataset            | Per-Tensor | Per-Channel |
> |----------------------------|------------|-------------|
> | LLaMA 3.1 8B - W2          |     6.29       |      6.29       |
> | LLaMA 3.1 8B - C4          |      9.67      |       9.68      |
> | LLaMA 3.1 8B - PTB         |       9.06     |       9.05      |
> | Mistral Nemo - W2          |     5.78       |       5.79      |
> | Mistral Nemo - C4          |      9.63      |       9.64      |
> | Mistral Nemo - PTB         |      8.70      |      8.72       |
> | Phi-4 - W2                 |      6.48      |      6.48       |
> | Phi-4 - C4                 |       12.10     |       12.10      |
> | Phi-4 - PTB                |       9.34     |       9.35      |
> | Mistral Small - W2         |      5.34      |       5.35      |
> | Mistral Small - C4         |       9.20     |       9.20      |
> | Mistral Small - PTB        |      8.06      |       8.06      |
>
> &nbsp;
> 2. **Regarding motivation to switch between FP8 and FP16 models**
> First and foremost, we would like to clarify that although the 95% confidence intervals of the two models overlap, this does not necessarily imply that the performance difference is negligible. We believe that the degradation observed in the FP8 model (e.g., a 16-point drop in mean performance) can still be considered meaningful, depending on the application context and perspective, and should be taken into account by practitioners when building production-quality systems.
> &nbsp;
> That said, we acknowledge that the performance degradation of a well-optimized FP8 model may not be dramatic enough to reach a universal consensus. In this range, subjective judgment can play a role. It is reasonable for some to consider the FP8 model sufficient, and the use of FP16 unnecessary. In response, we would like to emphasize that achieving such a level of FP8 optimization poses practical challenges in real-world production environments.
>
> &nbsp;
> Building a production-grade quantized model is not as simple as achieving good benchmark scores. The model must be robust across a wide range of inputs, without any edge cases where outputs become corrupted. Achieving such robustness often goes beyond applying well-established techniques like SmoothQuant; it typically requires iterative, heuristic tuning, which can be costly and time-consuming.
> &nbsp;
> For instance, the Llama-3 technical report, one of the most detailed public documents on FP8 model production, explains that while SmoothQuant serves as the foundation, additional heuristics such as selective quantization and hyperparameter tuning are essential. Without these refinements, the report notes that models may still produce corrupted outputs in certain scenarios, even if their benchmark scores are strong.
> &nbsp;
> This refinement process is extremely resource-intensive, despite the heuristics themselves seeming relatively simple. Each tuning step requires thorough testing to identify potential failure cases. The Llama-3 team reports conducting evaluation beyond benchmarks, including case-by-case analysis of over 100,000 responses using a reward model, an auxiliary model trained specifically for quality assessment. Training such a reward model, constructing large-scale evaluation datasets, and analyzing the results all demand substantial engineering effort.
> &nbsp;
> Not all organizations have the infrastructure or engineering capacity to support this level of validation. In such cases, relying solely on an FP8 model may be risky and co-deploying an FP16 model alongside the FP8 model can serve as a more robust and pragmatic strategy.

---

### Official Review · Reviewer_CdLp · 2025-07-01

**Clarity:** 3
**Significance:** 2
**Originality:** 3
**Rating:** 4
**Confidence:** 3

**Summary:**

The authors propose a method, NestedFP, to store both FP8 and FP16 in one 16-bit storage, which enables more memory efficient LLM serving compared to storing both models individually and more runtime-efficient than on-the-fly quantization. They provide a custom GEMM implementation built on CUTLASS.

**Questions:**

Why not show Table 2 as differences, as you did before for FP8 vs FP16 in Table 1?

**Ethical Concerns:**

["NO or VERY MINOR ethics concerns only"]

**Final Justification:**

Thank you for the clarifications.
The paper overall seems fine to me, but I am not confident enough to give it a higher score. I will maintain.

**Limitations:**

could be made more explicit into a small section. But they are detailed in the Checklist.

I am very much not an expert in this topic, so while this looks all fine to me, I can not actually gauge the work to its full depth.

**Quality:**

3

**Strengths And Weaknesses:**

Strengths

- reduced memory requirement
- good runtime speed-ups in FP8 with small cost in FP16
- very straightforward idea, and efficient implementation
- good experiments across multiple models to validate the general applicability
    - It wouldn’t hurt to go a little broader with the benchmarks though

Weaknesses

- 6.5% increase in GEMM runtime when using FP16
- This paper feels like it should go into a venue like USENIX ATC/OSDI or so. It does not really strike me as AI research. (Ofc, I will review under the assumption that NeurIPS is in fact the right venue)
- There is nothing really wrong with this paper, but I cannot gauge the impact. The method seems easy enough to adopt, but there are some significant tricks that had to be made to get the GEMM to be efficient, which sound like they could be overthrown by the next hardware change. And I wonder what experts think of the tradeoff between 3.9% increase end to end in FP16 for the benefit of less memory.

---

> ### Author Rebuttal · Authors · 2025-07-31
>
> Thank you for raising this thoughtful and important concern.
>
> 1. **Tradeoff Between Performance Overheads and Memory Savings**
> It is true that NestedFP incurs a small performance overhead (e.g., 3.9% throughput loss) in exchange for memory savings. In response to whether this tradeoff is worthwhile, we emphasize that in LLM inference, memory savings can often be reinvested to improve overall performance. For example, memory savings may enable larger batch sizes—thus increasing throughput—or reduce the frequency of KV cache evictions, which can become a significant bottleneck under memory pressure. While we did not incorporate such reinvestment scenarios in our evaluation in order to isolate the performance impact of kernel slowdown, we believe that, when leveraged, the performance gains enabled by NestedFP’s reduced memory footprint can more than offset the 3.9% throughput loss.
> &nbsp;
> To evaluate how memory savings of NestedFP can be invested to performance improvement, we ran controlled experiments using an H100 GPU with Azure traces (791 requests, 5 minutes) and compared NestedFP16 only deployment (no weight duplication, full KV cache capacity) with a dual deployment of FP16 and FP8 models (weight duplication, KV cache capacity reduced from 34GB → 12GB).
> &nbsp;
> Severe performance degradation was observed when co-deploying both models. P90 TTFT increased from 4.02s → 283.31s, total E2E latency more than doubled (from 300s → 627s), and throughput dropped by > 40%. This was mainly due to the limited batch size and frequent kv cache evictions.
> &nbsp;
> In short, the modest kernel slowdown is a worthwhile trade-off, as the resulting memory savings can be reinvested to mitigate key performance bottlenecks, ultimately outweighing the overhead.
> &nbsp;
> 2. **Regarding Presentation Clarity and Table Formatting** Thank you for your overall positive comments and thoughtful suggestions. As you pointed out, showing differences in Table 2—similar to Table 1—would improve clarity. We agree and will revise the table to explicitly highlight the differences in FP8 results as well.

---

### Official Review · Reviewer_R4or · 2025-07-04

**Clarity:** 3
**Significance:** 4
**Originality:** 3
**Rating:** 5
**Confidence:** 4

**Summary:**

NestedFP is a precision-adaptive weight format that splits every FP16 weight into two 8-bit halves. The upper half alone forms an E4M3-compatible tensor for fast FP8 inference, while both halves can be fused on-the-fly—via a custom CUTLASS GEMM kernel—to recover the exact FP16 value for accuracy-critical execution. Integrated into vLLM, the authors report up to 1.55× higher throughput in FP8 mode with negligible accuracy loss on Minerva-Math, MMLU-Pro and BBH, and only around 4% average overhead when running in FP16 mode, all without duplicating model weights in memory.

**Questions:**

- When reading only the upper 8 bits for FP8 inference, does it involve scattered loads that incur extra L1/L2 misses versus a contiguous FP8 checkpoint? Any empirical latency numbers compared with native FP8 tensors?
- Could NestedFP (or its variants) be generated directly from BF16?
- Given the next-gen Blackwell, can the same “nested” trick be extended to 4-bit? (I assume it's hard to directly transform it to MXFP4 or NVFP4; just want to know some insights.)
- The throughput report shows that NestedFP8 gains shrink with a large batch compared to vanilla FP8. Is this due to doubling weight-memory traffic (16B) or to reconstruction pipeline saturation?
- Could you report latency or TTFT.
- DFloat11 (https://arxiv.org/abs/2504.11651) is related, I'd love to see some comparison if possible.

**Ethical Concerns:**

["NO or VERY MINOR ethics concerns only"]

**Final Justification:**

NestedFP is an interesting work that splits every FP16 weight into two 8-bit halves and switch the floating point format on-the-fly. During rebuttal, the authors answered my questions and overall, this is a solid work with both good motivation and implementation.

**Limitations:**

See weaknesses.

**Quality:**

3

**Strengths And Weaknesses:**

Strengths:
- This method proposes an elegant nested representation that re-uses the unused FP16 MSB to embed an FP8 mantissa, enabling loss-free round-trips while keeping the checkpoint size unchanged.
- Thoughtful kernel design (SIMT fusion, three-stage pipeline) hides most reconstruction latency.
- System-level integration inside vLLM with comprehensive and realistic evaluation, and also comprehensive ablations on kernel.
- Zero extra RAM compared with pure FP16.

Weaknesses:
- Evaluation coverage, it would be better to have:
  - larger-scale models that are common in production
  - Reasoning-oriented checkpoints such as DeepSeek-R1, DS-Distiled-Llama, etc.
  - Datasets of coding or those that need longer generation, like LiveCodeBench
- Performance compromise: several datasets still show non-trivial accuracy degradation and, the throughput occasionally falls below vanilla ones
- A principled treatment of extremely large weights is still missing, limiting universality.
- Many open-weights today are stored in BF16. Converting BF16 → NestedFP adds an extra cast step—yet the paper does not evaluate this path.

---

> ### Author Rebuttal · Authors · 2025-07-31
>
> We sincerely thank the reviewer for the detailed and insightful feedback.
>
> 1. **Evaluation on Large Models/Reasoning Models** We evaluated the effectiveness of NestedFP on two 70B-scale models: LLaMA 3.1 70B, a general-purpose language model, and DeepSeek-R1-Distill-LLaMA-70B, which is specialized for reasoning tasks.
> &nbsp;
> - *Applicability*: We analyzed the weight distributions of LLaMA 3.1 70B and DeepSeek-R1-Distill-70B to assess the applicability of NestedFP across QKVO and MLP linear layers. For both LLaMA 3.1 70B and DeepSeek-R1-Distill-70B, over 93% of layers (523/560 and 521/560, respectively) satisfied the constraint. While the remaining layers are left in FP16, most layers can be converted to the NestedFP format. This demonstrates that NestedFP is applicable to both large-scale models and reasoning models.
> - *Accuracy*: We benchmarked both models on three tasks: Minerva MATH, BBH, and MMLU Pro. We compared FP16 results, FP8 baseline (per-token activation and per-tensor weight quantization), and our NestedFP8 format. Results on the table show that NestedFP8 consistently matches or outperforms the FP8 baseline across benchmarks, while maintaining accuracy very close to full-precision FP16. These results confirm that NestedFP8 is effective for both large-scale and reasoning models.
>
> | Model | Benchmark | FP16 | FP8 Baseline | NestedFP8 |
> |-------|-----------|------|-------------|-----------|
> | LLaMA 3.1 70B | Minerva | 0.3522 | 0.3526 | 0.3546 |
> |              | BBH     | 0.4956 | 0.4942 | 0.4933 |
> |              | MMLU Pro | 0.4751 | 0.4791 | 0.4745 |
> | DeepSeek-R1-Distill-70B | Minerva | 0.4534 | 0.4508 | 0.4468 |
> |                         | BBH     | 0.5365 | 0.5308 | 0.5356 |
> |                         | MMLU Pro | 0.4960 | 0.4897 | 0.4934 |
> - *Latency*:
> We measured inference latency on LLaMA 70B using H100 GPUs with tensor parallelism (TP=2). NestedFP8 achieves average x1.43 TTFT speedup over FP16 across different input lengths, while NestedFP16 incurs minimal overhead (<4% increase). These results confirm that the NestedFP format overhead remains negligible at large scale, while FP8 quantization provides significant speedup as GEMM operations dominate the computation.
>
> | Tokens (In, Out) | Metric | FP16  | FP8   | NestedFP16 | NestedFP8 |
> |------------------|--------|-------|-------|------------|-----------|
> | (1024, 32) | TTFT(ms) | 161.9 | 119.9 | 167.7 | 114.4 |
> |            | TPOT(ms) | 29.5  | 20.5  | 30.4  | 21.5  |
> | (2048, 32) | TTFT(ms) | 302.1 | 203.1 | 315.9 | 207.0 |
> |            | TPOT(ms) | 29.6  | 20.7  | 30.5  | 21.7  |
>
> 2. **Regarding the question on reading FP8 tensors in inference** In our implementation, the two 8-bit tensors (upper and lower) are stored as separate, fully contiguous FP8 checkpoints, each aligned in memory to prevent scattered access. During FP8 inference, only the upper tensor is loaded, and since it is accessed contiguously, it incurs no additional L1/L2 cache misses compared to native FP8 checkpoints.
> &nbsp;
> 3. **How to support BF16 weights?** If a model is released in BF16, it can be cast to FP16 during preprocessing and subsequently converted into NestedFPweights. The error introduced by this casting is usually negligible in practice. To demonstrate it, we conducted an element-wise comparison between the FP16-cast versions of LLaMA 3.1 8B and Qwen3 14B and their original BF16 counterparts and found out that not a single parameter in any layer showed a numerically meaningful deviation (≥ 1e-10 absolute or relative error). This negligible deviation is also reflected in the perplexity evaluations shown in the table below. Across all cases, casting to FP16 has minimal impact, supporting the use of FP16 as a preprocessing step for NestedFP.
>
> | Model          | Wikitext2 (BF16) | Wikitext2 (FP16) | C4 (BF16) | C4 (FP16) | PTB (BF16) | PTB (FP16) |
> |----------------|------------------|------------------|-----------|----------|-------------|------------|
> | Llama-3.1-8B   | 6.24             | 6.24             | 9.59      | 9.58     | 9.02        | 9.01       |
> | Mistral Nemo   | 5.75             | 5.74             | 9.58      | 9.57     | 8.66        | 8.66       |
> | Phi-4          | 6.46             | 6.46             | 12.07     | 12.07    | 9.32        | 9.32       |
> | Mistral Small  | 5.29             | 5.29             | 9.09      | 9.09     | 8.00        | 8.00       |
>
> 4. **Throughout reduction of NestedFP8 compared to Torch FP8**
> We used a custom CUTLASS kernel for FP8 evaluation. The observed performance gap is likely due to suboptimal kernel configuration (e.g., tile shapes), as we did not perform an extensive kernel tuning process—FP8 kernel optimization was not the primary focus of this work. However, based on your comment, we revisited this issue and found that there is no reason to use a custom kernel for NestedFP8; the same kernel used in the baseline can be applied. We will update our evaluation accordingly in future revisions and expect no  performance gap for FP8 GEMM.
> &nbsp;
> 5. **Latency (TTFT and TPOT)**
> We implemented a simple load-aware scheduler on the vLLM v1 engine. This scheduler selects NestedFP8 when the batch token count exceeds half of max_num_batched_tokens, and defaults to NestedFP16 otherwise—achieving adaptive precision switching without forecasting or user input. We conducted experiments on a single-node H100 80GB server using Mistral Small 24B as the test model Azure LLM trace.
>
> | Metrics       | FP16 | NestedFP |   |   |   |   |   |   |   |
> |---------------|------|----------|---|---|---|---|---|---|---|
> | p90 TTFT (ms) | 2420 | 890      |   |   |   |   |   |   |   |
> | p90 TPOT (ms) | 260  | 63       |   |   |   |   |   |   |   |
>
> 6. **Regarding DFloat11** DFloat11 leverages unused exponent bits in bfloat16 and introduces an efficient inference kernel. While both DFloat11 and NestedFP exploit unused exponent bits and employ custom kernels, their goals and effects differ. DFloat11 focuses on compressing model weights and using the saved memory to achieve performance gains, particularly in comparison to CPU offloading scenarios, without any loss in accuracy. In contrast, NestedFP does not reduce the actual model size. Instead, it supports both lossless FP16 inference and quantized FP8 inference to better meet SLO constraints. As mentioned earlier, even with a simple scheduler, NestedFP can deliver substantial latency improvements over static FP16-only serving.
> &nbsp;
> 7. **Regarding possible extensions to FP4** We appreciate the suggestions regarding both FP4 extensions. Since (1) FP4 requires more careful handling of scaling, and (2) FP8 offers very few unused exponent bits to exploit, extending similar ideas to FP4 is non-trivial—but still a direction worth exploring.

---

> > ### Comment · Reviewer_R4or · 2025-08-03
> >
> > I thank the authors for their response. Please incorporate these new experiments and discussions into the revised manuscript. Overall, this is an interesting paper, and I am maintaining my score.

---

> ### Author Response · Authors · 2025-08-04
>
> Thank you for the positive feedback and for maintaining your recommendation. We’ll incorporate the new experiments and discussions into the final version of the manuscript.

---

### Official Review · Reviewer_bPHS · 2025-07-05

**Clarity:** 3
**Significance:** 3
**Originality:** 3
**Rating:** 4
**Confidence:** 4

**Summary:**

This paper introduces **NestedFP**, a precision‐adaptive inference system that stores only a standard FP16 checkpoint which can be run entirely in FP8 or lossless FP16 on demand. It works by splitting each FP16 weight into two 8-bit tensors—an “upper” tensor that directly matches the E4M3 FP8 format and a “lower” tensor that, when fused back via bit-wise operations, exactly reconstructs the original FP16 value. A custom, branch-free CUTLASS GEMM kernel overlaps these reconstruction steps with tensor-core math, adding just ~6.5% compute overhead. Unlike prior approaches, there’s no extra memory for scales or metadata, and only a handful of out-of-range “exception” layers fall back to native FP16. Integrated into vLLM, NestedFP delivers up to 1.55× FP8 throughput (≤0.83% accuracy loss) and just 3.9% end-to-end overhead when running in FP16 across 8–24B models on benchmarks like MMLU and BBH.

**Questions:**

## Questions and Suggestions for the Authors

1. **Precision of NestedFP**
   - **Issue:** NestedFP is the same as FP15. In this case, is it necessary for us to use FP16?
   - **Request:** Please include an analysis or microbenchmarks for a more general model. Do we need 'E1' in the precision? I am not confident here we can save one bit for 'E1'.

2. **Memory Issue Justification**
   - **Issue:** The rationale for trading FP8 and FP16’s accuracy/throughput to save memory is not grounded in concrete use cases.
   - **Request:** Besides the Azure Worklog, give another case/description of inefficiency of deploying the fp8 and fp16 model at the same time. Also, can the author discuss whether this can be resolved on the scheduling level?

3. **Scalability to Very Large Models**
   - **Issue:** Evaluation is limited to small/medium models, leaving its behavior on 10 B+ parameter / Reasoning models unknown.
   - **Request:** Provide either (a) a cost projection (memory, compute, communication) for a llama 3.3 - 70B, Qwen-3, or DeepSeek R1.

4. **Detailed Accuracy Comparison**
   - **Issue:** The paper reports aggregate metrics but lacks a layer-wise accuracy breakdown against FP16. Also, like DeepSeek FP8 [https://github.com/deepseek-ai/DeepGEMM] has no performance drop if we only apply it on the attention layer, how does NestedFP work in this scenario?
   - **Request:** Add a table or plot showing per-layer or per-block relative error for NestedFP vs. FP16 vs DeepSeek FP8 on standard benchmarks.

5. **Scheduling Replicate**
  - **Issue** The paper is motivated by an Azure scheduling scenario, but I didn't see any scheduling analysis.
  - **Request** Use a simulated/real request flow in the author's vLLM implementation and show the improvement of TTFT / other performance metrics.

**Ethical Concerns:**

["NO or VERY MINOR ethics concerns only"]

**Final Justification:**

The technical concerns I had with this paper are mostly resolved. I will raise my point.

Please, include make sure the discussion in the final manuscript.

**Limitations:**

Partially yes. The authors need a more thorough discussion of potential scenarios where accuracy loss is acceptable and explicitly evaluate this trade-off. Additionally, elaboration on how NestedFP’s precision compromise affects specific critical reasoning would enhance the paper’s impact. There is not potential negative societal impact.

**Paper Formatting Concerns:**

No major formatting issues identified; the paper conforms to NeurIPS guidelines.

**Quality:**

3

**Strengths And Weaknesses:**

## Strengths

### Quality
- **Practical Implementation and Solid Evaluation** The author has a super solid implementation on vLLM and A branch-free CUTLASS GEMM kernel hides the bit-manipulation overhead (< 6.5 %).

### Clarity
- **Illustrative figures.** Diagrams of the dual-tensor layout and pipelined kernel aid understanding of low-level mechanics.

### Significance
- **Per-iteration precision switching.** Enables dynamic adjustment to workload between FP16 and FP8.

### Originality
- **Novel dual-8-bit decomposition.** Re-purposes the unused FP16 exponent bit to encode an FP8 tensor, yet still reconstructs exact FP16 values.

---

## Weaknesses

### Quality
- **Weight distributions requirement** The whole idea is built on the motivation that the majority of weight values have absolute magnitudes less than or equal to 1.75. If this condition not exist, the nestedfp cannot be achieved.

### Clarity
- **Scattered narrative.** The introduction over-emphasizes FP8 background, delaying presentation of the core mechanism.
- **Confusing Motivation** It needs a lot of effort to understand Figure 1 (a). The author can show the figure in 0/1 => under / above the average usage.

### Significance
- **Exception-layer overhead.** “Few” fallback layers are reported but not analyzed per model; impact on larger models is unclear to me.
- **Necessity analysis.** Azure Worklog is not enough for me. I want to see if this issue exists in the LLM production environment.
- **NestedFP vs FP16** If we use this way to restore fp16 back, the weakness of NestedFP will bring performance and accuracy drop.

### Originality
- **Literature contrast.** Similar on-the-fly quant-dequant methods exist; a deeper comparison to dynamic quantization kernels is needed.

---

> ### Author Rebuttal · Authors · 2025-07-31
>
> We sincerely thank the reviewer for the thoughtful and constructive feedback.
> &nbsp;
> 1. **Precision of NestedFP** FP16 is a standard data type widely supported by hardware accelerators, including tensor cores. While it is possible to store weights using a 15-bit representation (e.g., by omitting one exponent bit as in FP15), doing so requires custom kernels to reconstruct valid FP16 values at runtime—just as NestedFP does. The key insight is that we're not creating FP15 for its own sake, but rather enabling dual-precision execution from a single representation. To evaluate the general applicability of NestedFP, we provide a detailed analysis of 14 models in Appendix F. These results suggest that a wide range of state-of-the-art models can benefit from NestedFP.
> &nbsp;
> 2. **Memory Issue Justification:Why We Should Save Memory**
> First and foremost, deploying FP8 and FP16 models separately increases memory usage by approximately 1.5× compared to deploying a single FP16 model, which can limit the maximum model size that fits within GPU memory.
> &nbsp;
> Second, in LLM inference, memory capacity is not merely a resource constraint—it has direct implications on performance. Memory savings can often be reinvested to improve system efficiency. For example, memory savings may enable larger batch sizes—thus increasing throughput—or reduce the frequency of KV cache evictions, which can become a significant bottleneck under memory pressure. While we did not incorporate such reinvestment scenarios in our evaluation in order to isolate the performance impact of kernel slowdown, we believe that, when leveraged, the performance gains enabled by NestedFP’s reduced memory footprint can more than offset the throughput loss caused by kernel slowdown (i.e., 3.9%).
> &nbsp;
> To evaluate how memory savings of NestedFP can be invested to performance improvement, we ran controlled experiments using an H100 GPU with Azure traces (791 requests, 5 minutes) and compared NestedFP16 only deployment (no weight duplication, full KV cache capacity) with a dual deployment of FP16 and FP8 models (weight duplication, KV cache capacity reduced from 34GB → 12GB).  Severe performance degradation was observed when co-deploying both models. P90 TTFT increased from 4.02s → 283.31s, total E2E latency more than doubled (from 300s → 627s), and throughput dropped by > 40%. This was mainly due to limited batch sizes and frequent KV cache evictions.
> &nbsp;
> In short, the modest kernel slowdown is a worthwhile trade-off, as the resulting memory savings can be reinvested to mitigate key performance bottlenecks, ultimately outweighing the overhead.
> &nbsp;
> 3. **Evaluation on Large Models/Reasoning Models** We evaluated the effectiveness of NestedFP on two 70B-scale models: LLaMA 3.1 70B, a general-purpose language model, and DeepSeek-R1-Distill-LLaMA-70B, which is specialized for reasoning tasks.
> &nbsp;
> - *Applicability*: We analyzed the weight distributions of LLaMA 3.1 70B and DeepSeek-R1-Distill-70B to assess the applicability of NestedFP across QKVO and MLP linear layers. For both LLaMA 3.1 70B and DeepSeek-R1-Distill-70B, over 93% of layers (523/560 and 521/560, respectively) satisfied the constraint. While the remaining layers are left in FP16, most layers can be converted to the NestedFP format. This demonstrates that NestedFP is applicable to both large-scale models and reasoning models.
> - *Accuracy*: We benchmarked both models on three tasks: Minerva MATH, BBH, and MMLU Pro. We compared FP16 results, FP8 baseline (per-token activation and per-tensor weight quantization), and our NestedFP8 format. Results on the table show that NestedFP8 consistently matches or outperforms the FP8 baseline across benchmarks, while maintaining accuracy very close to full-precision FP16. These results confirm that NestedFP8 is effective for both large-scale and reasoning models.
>
> | Model | Benchmark | FP16 | FP8 Baseline | NestedFP8 |
> |-------|-----------|------|-------------|-----------|
> | LLaMA 3.1 70B | Minerva | 0.3522 | 0.3526 | 0.3546 |
> |              | BBH     | 0.4956 | 0.4942 | 0.4933 |
> |              | MMLU Pro | 0.4751 | 0.4791 | 0.4745 |
> | DeepSeek-R1-Distill-70B | Minerva | 0.4534 | 0.4508 | 0.4468 |
> |                         | BBH     | 0.5365 | 0.5308 | 0.5356 |
> |                         | MMLU Pro | 0.4960 | 0.4897 | 0.4934 |
> - *Latency*:
> We measured inference latency on LLaMA 70B using H100 GPUs with tensor parallelism (TP=2). NestedFP8 achieves average x1.43 TTFT speedup over FP16 across different input lengths, while NestedFP16 incurs minimal overhead (<4% increase). These results confirm that the NestedFP format overhead remains negligible at large scale, while FP8 quantization provides significant speedup as GEMM operations dominate the computation.
>
> | Tokens (In, Out) | Metric | FP16  | FP8   | NESTEDFP16 | NESTEDFP8 |
> |------------------|--------|-------|-------|------------|-----------|
> | (1024, 32) | TTFT(ms) | 161.9 | 119.9 | 167.7 | 114.4 |
> |            | TPOT(ms) | 29.5  | 20.5  | 30.4  | 21.5  |
> | (2048, 32) | TTFT(ms) | 302.1 | 203.1 | 315.9 | 207.0 |
> |            | TPOT(ms) | 29.6  | 20.7  | 30.5  | 21.7  |
>
> 4. **Detailed Accuracy Comparison** To perform a per-layer analysis of the decoder, we quantized all QKVO and MLP FP16 linear layers into FP8 and measured the relative error across decoder layers using the Pile training dataset. Results for selected layers (L10, L20, L30, L40) and the full-model average show that NestedFP8 generally yields slightly higher error than DeepSeek FP8 due to its coarser quantization. As NestedFP8 applies only to linear layers, attention-layer quantization schemes like DeepSeek FP8 can still be applied independently without conflict.
>
> | Model          | Method        | L10    | L20    | L30    | L40    | Avg.   |
> |:--------------|:--------------|:------:|:------:|:------:|:------:|:-----:|
> | Llama 3.1 8B   | DeepSeek FP8  | 0.088  | 0.075  | 0.080  |   -    | 0.077 |
> |                | NestedFP8     | 0.112  | 0.090  | 0.096  |   -    | 0.095 |
> | Mistral Nemo   | DeepSeek FP8  | 0.079  | 0.081  | 0.075  | 0.084  | 0.076 |
> |                | NestedFP8     | 0.097  | 0.097  | 0.089  | 0.099  | 0.094 |
> | Phi-4          | DeepSeek FP8  | 0.056  | 0.072  | 0.058  | 0.063  | 0.059 |
> |                | NestedFP8     | 0.064  | 0.082  | 0.066  | 0.069  | 0.066 |
> | Mistral Small  | DeepSeek FP8  | 0.055  | 0.070  | 0.062  | 0.067  | 0.060 |
> |                | NestedFP8     | 0.072  | 0.086  | 0.077  | 0.092  | 0.077 |
>
> 5. **Scheduling Replicate**
> First and foremost, we would like to emphasize that the primary goal of our work is to enable per-iteration precision switching (every 10–100 ms) between FP8 and FP16 without model duplication. NestedFP is scheduler-agnostic and can be seamlessly integrated with any LLM scheduling policy. While the design of such schedulers is beyond the scope of this paper, we consider developing effective scheduling strategies for NestedFP a promising direction for future research.
> &nbsp;
> Nevertheless, to provide a rough estimate of the potential performance benefits in a realistic serving scenario, we implemented a simple load-aware scheduler on the vLLM v1 engine. This scheduler selects NestedFP8 when the batch token count exceeds half of max_num_batched_tokens, and defaults to NestedFP16 otherwise—achieving adaptive precision switching without forecasting or user input. Note that this is a very naive policy, and many optimization opportunities remain on the table, such as incorporating per-query SLO constraints, KV cache utilization, or more sophisticated load prediction mechanisms.
> &nbsp;
> We conducted experiments on a single-node H100 80GB server using Mistral Small 24B as the test model Azure LLM trace. In addition to TTFT and TPOT, we evaluated SLO attainment by defining two SLO classes following the latency analysis framework suggested by SOLA (MLSys'25). Specifically, we set tight SLO as ≤ 5× the average single-request latency and loose SLO as ≤ 10× the average single-request latency, where the single-request latency was measured in our experiment environments.
> &nbsp;
> Even with simple policy, being able to dynamically switch between FP8 and FP16 led to dramatic latency improvements over static FP16-only serving:  **p90 TTFT decreased from 2.42s → 0.89s (x0.36), p90 TPOT decreased from 0.26s → 0.063s (x0.24), tight SLO attainment increased from 49.9% → 95.1% (x1.90) and loose SLO attainment increased from 89.5% → 100% (x1.12).**
> &nbsp;
> These results demonstrate the effectiveness of token-level precision switching, highlighting the motivation behind NestedFP, which is to enable it in a memory-efficient manner.

---

> > ### Comment · Reviewer_bPHS · 2025-08-06
> >
> > Thanks for additional experiment.
> >
> > 1-4 answers my concerns impressively. But I do still have concerns for 5.
> >
> > > NestedFP is scheduler-agnostic and can be seamlessly integrated with any LLM scheduling policy
> >
> > LLM scheduling policy will consider model size, therefore NestedFP definitely will affect scheduling efficiency. I do want to see more experiments based on this for simulated worklog in multi-node setting (similar to the paper SOLA mentioned by the author).

---

> ### Author Response · Authors · 2025-08-07
>
> First, we would like to clarify that NestedFP does not incur any additional memory overhead compared to a standard FP16-only deployment. While the simple scheduling policy used in our proof-of-concept experiment bases decisions solely on system load, it is true that some scheduling policies may take memory size into account for various reasons. However, even under such policies, NestedFP does not lead to any scheduling inefficiencies related to memory usage.
>
> In response to your request for additional results, we conducted the same experiment described in Answer #5, this time using a 4×H100 distributed setup with the Llama 70B model.
> The results under the highest tested load (7.47 req/s) show that our approach significantly outperforms the FP16-only baseline:
> p90 TTFT decreased from 12.06s → 1.66s (x0.14).
> Tight SLO attainment increased from 0% → 50.3%.
> Loose SLO attainment increased from 5.9% → 85.9% (x14.5).
> The full results across all tested request rates are summarized below:
> | Request Rate | Method | Tight SLO | Loose SLO | p90 TTFT(s) |
> | :--- | :--- | :--- | :--- | :--- |
> | **7.47 req/s** | FP16 | 0.00% | 5.90% | 12.06 |
> | | **NestedFP** | **50.30%** | **85.90%** | **1.66** |
> | **5.03 req/s** | FP16 | 43.40% | 82.90% | 1.79 |
> | | **NestedFP** | **78.30%** | **96.60%** | **0.91** |
> | **3.79 req/s** | FP16 | 62.40% | 91.90% | 1.29 |
> | | **NestedFP** | **82.60%** | **98.00%** | **0.79** |
> | **1.99 req/s** | FP16 | 74.70% | 95.40% | 0.93 |
> | | **NestedFP** | **88.90%** | **99.60%** | **0.59** |
>
> These results confirm that our approach—choosing between NestedFP8 and NestedFP16 based on system load—yields consistent, significant gains even for a large model in a distributed setting. We believe this addresses your final concern and will incorporate the findings into our manuscript. Thank you.

---

> > ### Comment · Reviewer_bPHS · 2025-08-08
> >
> > The technical concerns I had with this paper are mostly resolved. I will raise my point.
> >
> > Please, include our discussion in the final manuscript.

---

> > > ### Author Response · Authors · 2025-08-09
> > >
> > > We appreciate your valuable feedback and are glad that the technical concerns have been addressed. We will make sure to incorporate the relevant points from our discussion into the final manuscript.

---

### Note · Authors · 2025-08-13

During the author–reviewer discussion period, reviewers raised several concerns, most of which were addressed through additional experiments and analyses, as explicitly acknowledged by Reviewers bPHS and R4or. In summary, these concerns included: the applicability of NestedFP to general, large-scale, and reasoning models; its advantages over co-deploying separate FP8 and FP16 models; layer-wise relative error analysis of NestedFP8 versus DeepSeek FP8; improvements in single- and multi-GPU TTFT and SLO attainment; comparisons with torch FP8; support for BF16 checkpoints; differences from DFloat11; potential FP4 extensions; and perplexity evaluations.
&nbsp;
The only point that did not appear to reach full author–reviewer agreement was raised by Reviewer B4rE: *“Doesn’t the SOTA FP8 model work as well as the FP16 model? Why keep both?”* As a final remark, we reiterate our response succinctly: while a well-optimized FP8 model may match FP16 model, achieving such quality is technically challenging and often beyond the reach of many practitioners. In these cases, FP8 models may underperform relative to FP16, making it important to maintain both versions. A full response is provided in our last comment to Reviewer B4rE.
&nbsp;
We sincerely thank the reviewers and ACs for their constructive feedback, which we will incorporate into the revised version.

---

### Decision · Program_Chairs · 2025-09-17

**Decision:**

Accept (poster)

**Comment:**

The paper introduces NestedFP, a dual-precision weight representation that encodes each FP16 weight as two 8-bit parts so that a single 16-bit checkpoint can execute either as fast FP8 (using only the upper 8-bit tensor) or as exact FP16 (by fusing both parts on the fly). A custom, branch-free CUTLASS GEMM overlaps reconstruction with tensor-core compute and is integrated into vLLM. Reported results show up to 1.55× higher throughput in FP8 mode with negligible accuracy loss on reasoning benchmarks, and ~3.9–4.5% average end-to-end overhead in FP16 mode without duplicating weights in memory.

Strength
1) An elegant nested encoding lets one FP16 checkpoint serve both FP8 and FP16 without extra scales/metadata, avoiding memory bloat common in co-deploying separate FP8/FP16 models. Exact FP16 recovery is guaranteed.
2) CUTLASS kernel pipelines reconstruction to hide latency; authors show cache-friendly FP8 layout (no scattered loads) and integrate everything into vLLM, enabling end-to-end serving experiments.
3) The paper goes beyond GEMMs: SLO-aware batching/precision scheduling reduces TTFT and improves SLO attainment under Azure traces, including multi-GPU settings.
4) Results span multiple model families/sizes (incl. 70B) and compare FP8 error against a strong baseline (DeepSeek-FP8)

Weakness
1) The method relies on most weights having |w|<=1.75 to fit E4M3; layers violating this become “exceptions” that fall back to FP16. The paper shows only a handful of such layers, but scenarios with heavier tails (other checkpoints/training recipes) could reduce FP8 coverage.
2) While modest (~3.9-4.5%), it may still matter for latency-sensitive paths; the kernel optimizations are hardware-specific and could shift with future architectures.
3) FP8 relative error is “close but slightly higher” than DeepSeek-FP8 in some layers; the paper argues it’s negligible at model level, but a deeper analysis of pathological cases (reasoning/coding long-context) would be valuable.

Recommendation:
Unifying FP8/FP16 from a single checkpoint, while preserving exact FP16—solves a real deployment problem. The combination of kernel work, vLLM integration, and SLO-aware scheduling demonstrates tangible end-to-end benefits (TTFT/TPOT & throughput) beyond microbenchmarks. I stop short of spotlight: while impactful, parts of the empirical story (BF16 conversion path, edge cases with heavy-tailed weights, deeper long-context/coding analyses) could be more comprehensive.

Net assessment after rebuttal:  Most substantive concerns (applicability to larger models, serving metrics beyond GEMM, BF16 path clarity, comparisons to FP8 baselines, per-layer error) were addressed with additional results and discussion; only the “why keep FP16 if FP8 can suffice” question remained partially debated. Given the single-checkpoint deployment win, strong system integration, and measured end-to-end benefits, I recommend Accept (Poster).